# Pharmacokinetic of Cefiderocol in Critically Ill Patients Receiving Renal Replacement Therapy: A Case Series

**DOI:** 10.3390/antibiotics11121830

**Published:** 2022-12-16

**Authors:** Simone Mornese Pinna, Silvia Corcione, Amedeo De Nicolò, Giorgia Montrucchio, Silvia Scabini, Davide Vita, Ilaria De Benedetto, Tommaso Lupia, Jacopo Mula, Giovanni Di Perri, Antonio D’Avolio, Francesco Giuseppe De Rosa

**Affiliations:** 1Department of Medical Sciences, Infectious Diseases, University of Turin, 10126 Turin, Italy; 2School of Medicine, Tufts University, Boston, MA 02111, USA; 3Laboratory of Clinical Pharmacology and Pharmacogenetics, Department of Medical Sciences, University of Turin, Amedeo di Savoia Hospital, 10126 Turin, Italy; 4Department of Anesthesia, Intensive Care and Emergency, Citta della Salute e della Scienza Hospital, University of Turin, 10124 Turin, Italy; 5ASL Asti, Cardinal Massaia Hospital, 14100 Asti, Italy; 6Unit of Infectious Diseases, Department of Medical Sciences, University of Turin, Amedeo di Savoia Hospital, 10149 Turin, Italy

**Keywords:** cefiderocol, pharmacokinetics, renal replacement therapy, CRRT, CVVH, therapeutic drug monitoring

## Abstract

Background: Cefiderocol is a novel parenteral siderophore cephalosporin, demonstrating enhanced activity against multidrug-resistant (MDR) Gram-negative bacteria and difficult-to-treat *Acinetobacter baumannii* (DTR-AB). Plasma-free trough concentration (*f*C_trough_) over the minimum inhibitory concentration (MIC) was reported as the best pharmacokinetic parameter to describe the microbiological efficacy of cefiderocol. Materials and methods: We retrospectively described the pharmacokinetic and pharmacodynamic profile of three critically ill patients admitted to the intensive care unit, receiving cefiderocol under compassionate use to treat severe DTR-AB infections while undergoing continuous venovenous haemofiltration. Cefiderocol was administrated at a dosage of 2 g every 8 h infused over 3 h. Therapeutic drug monitoring (TDM) was assessed at the steady state. Cthrough was evaluated by assuming a plasma protein binding of 58.0%. The fCmin/MIC was calculated assuming a cefiderocol MIC equal to the PK-PD breakpoint of susceptibility ≤ 2. The association between the PK/PD parameters and microbiological outcome was assessed. Results: *f*C_trough_/MIC were >12 in 2 patients and 2.9 in the 1 who rapidly recovered from renal failure. Microbiological cure occurred in 3/3 of patients. None of the 3 patients died within 30 days. Conclusions: A cefiderocol dosage of 2 g q8 h in critically ill patients with AKI undergoing CVVH may bring about a very high plasma concentration, corresponding to essentially 100% free time over the MIC for DTR-AB.

## 1. Introduction

Non-fermenting Gram-negative bacilli (NFGNB) have emerged as a major cause of healthcare-associated infection in critically ill patients worldwide [1]. Among NFGNBs, the spread of difficult-to-treat (DTR) *Acinetobacter baumannii* (AB) and *Pseudomonas aeruginosa* (PA), characterized by high resistance to fluoroquinolone, β-lactams, including β-lactam/β-lactamase inhibitor combinations and carbapenems, constitutes an important burden for healthcare systems and a major clinical challenge, leading to high mortality in intensive care units.

Cefiderocol is a novel chlorocatechol-substituted siderophore cephalosporin known to form an iron-chelating complex to gain access to Gram-negative membranes, showing enhanced in vitro activity against DTR Enterobacterales and NFGNBs, including strains harboring metallo-β-enzymes [2]. An early and appropriate antimicrobial treatment demonstrated a reduction in mortality [3,4]. Optimization of the antibiotic dose–effect relationship in critically ill patients is crucial to achieving the therapeutic window, avoiding subtherapeutic or toxic concentration. However, other than the intrinsic characteristic of the molecule, even pathophysiological changes related to critical illness—such as modification of the volume of distribution and binding protein, hepatic dysfunction, and systemic inflammatory response syndrome-related changes—can affect antibiotic dosing in this population [5]. Critically ill patients often have rapidly fluctuating renal function, leading to acute kidney injury (AKI) requiring forms of renal replacement therapy (RRT). Moreover, RRT systems are known to affect the pharmacokinetic parameters of antimicrobials, adding further consideration to the complexity of critically ill patients [6,7,8].

The pharmacokinetic cefiderocol was recently evaluated in patients with renal dysfunction, but dosing in patients receiving CVVH is mainly extrapolated from cefepime pharmacokinetics (PK) because of the similarity in molecular weights and protein binding between cefepime and cefiderocol [9,10,11]. Here, we report for the first time the pharmacokinetic cefiderocol in three critically ill patients undergoing CVVH with severe infections due to DTR-AB.

## 2. Results

Three patients were evaluated in this study. Their SOFA scores ranged from 10 to 13, and their APACHE II scores ranged from 8 to 21. At the onset of cefiderocol treatment, the patients were critically ill, mechanically ventilated and receiving vasopressor support due to septic shock. Due to acute kidney failure, renal replacement therapy was started with CVVH. Adjunctive filters were employed in our patients to remove hydrophobic substances, such as inflammatory cytokines. In two patients, residual diuresis persisted at the time of CVVH support.

### 2.1. Patient 1

A 56-year-old man with a medical history significant for chronic obstructive pulmonary disease and bullous emphysema disease was admitted in July 2021 for a bilateral lung transplant. His postoperative course was complicated by respiratory failure due to grade III primary graft dysfunction and enteric colonization by DTR-AB. On day 7, he developed sepsis due to DTR-AB ventilator-associated pneumonia (VAP). The initial treatment with high dosage ampicillin/sulbactam, plus iv colistin and prolonged vasopressor, resulted in acute renal failure, requiring the initiation of CVVH.

CVVH was conducted with a heparinized circuit with the following setting: subtractive flow −150 mL/h BI −400 cc. Effluent flow rate 24 h 2300 mL on the day of Pk sampling.

Based on antimicrobial susceptibility, antibiotic treatment was changed to cefiderocol 2 g every 8 h. PK parameters were as follows: Ctrough 57.95 mg/L, Cmax 104.94 mg/L, t1/2 7.6 h and AUC0–8 h 643.90 mg/L*h. Considering 58% plasma protein binding, the plasma free trough concentration (fC_trough_) was 24.34 mg/L, 12.2-fold higher than the adopted EUCAST interpretative criteria for cefiderocol against AB of 2 mg/L [12,13]. The patient clinically improved, reducing systemic inflammatory markers and the vasopressor dosing. A quantitative culture on a lower respiratory sample after antibiotic treatment was negative. Renal function improved, and the patient was weaned from CVVH. During the subsequent intensive care unit (ICU) stay, he developed different episodes of bloodstream infections due to methicillin-resistant S. epidermidis and E. faecalis, and invasive pulmonary aspergillosis. On day 62 since ICU admission, the patient developed an episode of VAP due to *P. aeruginosa* and *K. pneumoniae* with refractory septic shock, despite maximal vasopressor support, and on day 72 he died.

### 2.2. Patient 2

A 71-year-old man with no medical history was admitted to the emergency department of a peripheral hospital because of shortness of breath. About eight days before, he tested positive for SARS-CoV−2. The chest scan revealed extensive bilateral ground glass opacities. Oxygen via nasal cannula was started, but acute respiratory failure developed after one day, requiring ventilation with HCPAP. After five days since hospital admission, following repeated unsuccessful attempts at prone positioning, the clinical condition suddenly deteriorated, and he was moved to our hospital for intubation and ICU admission due to acute respiratory distress syndrome related to SARS-CoV-2 pneumonia. At admission, volume-controlled ventilation was started with cycles of prone positioning. IV dexamethasone and an empiric antibiotic course with ceftriaxone were started. A rectal swab was positive for DTR-AB. Six days after admission, oxygen saturation levels improved, and respiratory weaning from mechanical ventilation was started. However, in the eight days of ICU, he developed fever, neutrophil leukocytosis and a new onset of respiratory failure requiring new intubation. Blood culture results were positive for DTR-AB, E. faecalis and E. faecium, and a bronchoalveolar lavage (BAL) culture was positive for DTR-AB > 100,000 CFU/mL. He eventually required norepinephrine to maintain hemodynamic stability and CVVH for AKI.

CVVH was conducted with a heparinized circuit with the following setting: subtractive flow rate 200 mL/min, dialysate flow rate 200 mL/min. Effluent flow rate 24 h 2600 mL on the day of PK sampling.

Given the acute renal failure and the critical illness, colistin, the only choice based on antibiogram susceptibility, was considered a suboptimal choice. Consequently, antibiotic treatment was changed to cefiderocol 2 g every 8 h, and iv fosfomycin 4 g every 6 h was added for its intrinsic activity against Enterococcus spp., favorable PK/PD behavior in lung tissue and synergistic effect when used within a combination treatment for the treatment of MDR organisms. There are encouraging experiences with the use of a cefiderocol combination regimen in the treatment of MDR organisms, including DTR-AB [14]. Cefiderocol PK parameters were as follows: Ctrough 14.17 mg/L, Cmax 64.87 mg/L, t1/2 2.0 h and AUC0–8 h 272.11 mg/L*h. Considering 58% plasma protein binding, the *f*C_trough_ was 5.95 mg/L, 3.0-fold higher than the adopted EUCAST interpretative criteria for cefiderocol against DTR-AB of 2 mg/L [12,13].

Eventually, the patient’s condition stabilized, with rapid resolution of the septic event. Subsequent microbiological BAL cultures were negative, and he was started on a slow respiratory weaning from mechanical ventilation. He recovered his renal function and CVVH was eventually discontinued.

### 2.3. Patient 3

A 45-year-old man with a past medical history of obesity and hypertension was admitted to our hospital for fever, dyspnea and diarrhea over the previous 5 days. He had a positive nasopharyngeal swab for SARS-CoV-2 since the onset of symptoms. At admission, he had hypoxaemic respiratory failure with P/F < 100 requiring non-invasive ventilation support. He deteriorated on day 2, with worsening dyspnea and hypotension, requiring ICU admission, invasive mechanical ventilation and hemodynamic support. A CT scan revealed a 21 cm cross-sectional diameter pneumomediastinum, subcutaneous emphysema and cardiac herniation. A chest tube was inserted, with progressive resolution of pneumomediastinum and hypotension. On day 12 since orotracheal intubation, he developed fever. Norephinephrine and fluid support were used to maintain blood pressure, and broad-spectrum antibiotics with piperacillin/tazobactam and vancomycin were started. A rectal swab was positive for carbapenemase-producing *K. pneumoniae*. Hypovolemic-induced AKI, which was refractory to conservative treatment, required the introduction of continuous CVVH. Blood cultures were positive for DTR-AB. Concomitantly, a tracheal culture specimen was positive for methicillin-resistant staphylococcus aureus and DTR-AB. Antibiotic treatment was changed to aerosolized colistin 1 MU every 8 h, i.v. linezolid 600 mg every 12 h and cefiderocol 2 g every 8 h. Cefiderocol PK parameters were as follows: Ctrough 71.66 mg/L, Cmax 117.40 ng/mL, t1/2 9.6 h and AUC0–8 h 772.66 ng/mL*h. Considering 58% plasma protein binding, the *f*C_trough_ was 30.10 mg/L, resulting in 15.0-fold higher than the adopted EUCAST interpretative criteria for cefiderocol against AB of 2 mg/L [13]. He developed a significant maculopapular rash on day 5 since the antibiotic change. Systemic corticosteroid and antihistamine medication were introduced with the suspicion of a drug-related adverse event. Clinical conditions improved, complete hemodynamic support weaning was performed and, after a seven-day course, antibiotic treatment was discontinued due to the progression of skin rash. Blood cultures and BAL cultures after antibiotic treatment showed no microorganisms. His respiratory function recovered, and a complete respiratory weaning was performed. Even the skin rash improved, and after 14 days of renal replacement treatment, CVVH was finally discontinued, and the patient was moved to a medical ward.

We found a *C*_through_ of ID1 (57.945 mg/L) and ID3 (71.664 mg/L) support, respectively, 4 and 5 times higher than ID2 (14.168 mg/L). The protein binding of cefiderocol ranges between 40−60% [15]. With an estimated unbound fraction of cefiderocol of 0.42, assumed from previous studies [9], we found *f*C_trough_ of 24.34 mg/L in ID1, 5.95 mg/L in ID2 and 30.10 mg/L in ID3. In all three patients, the *f*C_trough_ was far above the MIC of DTR-AB, but mainly in ID1 and ID3, compared to ID2, in which renal function rapidly improved.

## 3. Discussion

PK models are essential for antimicrobial dose optimization in special populations, such as patients undergoing RRT. It is well known that critically ill patients receiving CVVH are at risk of suboptimal dosing of antimicrobials and worse outcomes [16]. AKI complicates the clinical course of patients admitted to the ICU who develop sepsis or septic shock, and it is associated with increased morbidity and mortality [17]. Furthermore, patients suffering from an excessive pro-cytokine release could combine cytokine absorbers with CVVH in order to contain the systemic inflammatory response during sepsis [18,19].

To our knowledge, there is limited clinical data for cefiderocol PK in patients undergoing CVVH [9,10,11]. In this case series, we provide data on the PK of cefiderocol as salvage therapy in three critically ill patients on CVVH and cytokine absorbers while affected by severe DTR-AB infections with limited treatment options (Table 1). The manufacturer’s dosing recommendations for cefiderocol in patients receiving CVVH range from 1.5 g q12 h to 2 g q8 h. Considering the severity of illness of our patients and the PK changes in hydrophilic antibiotics occurring during sepsis and septic shock [20], we administered the maximum dose suggested for patients receiving renal replacement therapy of 2 g q8 h.

Ctrough of ID1 (57.945 mg/L) and ID3 (71.664 mg/L) were, respectively, 4 and 5 times higher than ID2 (14.168 mg/L). The protein binding of cefiderocol ranges between 40−60% [16]. With an estimated unbound fraction of cefiderocol of 0.42, assumed from previous studies [9], we found *f*C_trough_ of 24.34 mg/L in ID1, 5.95 mg/L in ID2 and 30.10 mg/L in ID3. In all three patients, the *f*C_trough_ were far above the MIC of DTR-AB, but mainly in ID1 and ID3, compared to ID2, in which the renal function rapidly improved.

In ID1 and ID2, residual renal function was present, while ID3 presented no urine output. Notably, at the time of PK analysis, ID2 had a resolving pattern of acute renal injury (AKI), and CVVH was rapidly discontinued two days after PK analysis, while the other two patients underwent prolonged renal support. This clinical pattern in patient ID2 perfectly explains the reduced t1/2 and, therefore, lower cefiderocol plasma concentration compared to the other two patients. These data indicate that the dosing of 2 g q8 h of cefiderocol suggested for patients receiving CVVH provides a very high plasma level in critically ill patients with severe infections.

Recently, Kobic et al. reported a similar *f*C_trough_ 31.6 mg/L in a case report describing the PK of a cefiderocol standard regimen in a patient undergoing CVVH for bloodstream infection by *P. aeruginosa* [11]. Interestingly, in ID2, the observed t1/2 was within the normal range described for patients with normal renal function (about 2 h). This confirms that ID2 was rapidly recovering from AKI. Moreover, all these results are particularly concordant with the data described by Katsube et al., reporting a mean t1/2 of 2.8 h and 9.6 h in normal renal function and hemodialysis, respectively [9].

A recent study on pigs did not demonstrate a significantly higher clearance of β-lactams with the application of the cytokine absorber alone, but it has been suggested that the combination of cytokine absorbers and CVVH resulted in an augmented clearance of meropenem in vitro [21,22]. More recently, Konig et al. demonstrated a cefiderocol free plasma concentration higher than the MIC in five patients with septic shock due to multidrug-resistant Gram-negative bacteria treated with different doses of cefiderocol during CVVH [23]. However, the application of CytoSorb^®^ to CVVH in 1/5 patients resulted in reduced cefiderocol plasma concentration compared to patients treated with only CVVH [23].

Interestingly, during CVVH, it was estimated that the effluent flow rate was the only significant variable influencing dosing optimization for cefideroco [15]. The effluent flow rate of patients in our study ranged from 2300 mL/h to 3600 mL/h. If we applied these effluent flow rates to the described model, ID1 and ID2 would need a 2 g q12 h dosage, while ID3, which had the highest plasma concentration, would require 1.5 g q8 h. It seems that the PK of antibiotics in critically ill patients undergoing CVVH provides a complex in vivo model, in which the CVVH configuration represents one of the critical goals to achieve in order to provide optimal antibiotic dosing in this population. However, other factors, such as the severity of the illness, should be considered.

Nevertheless, it appears evident that in our series, the cefiderocol dosage of 2 g q8 h resulted in high t1/2 in patients undergoing CVVH.

As for other β-lactams, % fT > MIC is the best PK/PD parameter for predicting the probability of PK-PD target attainment during treatment with cefiderocol. Based on murine PK/PD models, >75% fT > MIC has >90% probability of target attainment for susceptible organisms exhibiting MIC≤ 4 mg/L, including DTR microorganisms [24]. EUCAST MIC distributions show a modal MIC of cefiderocol against *A. baumannii* of 0.06 mg/L, and only 4.8% of 707 isolates showed a MIC ≥ 2.

At the time of this study, the e-test for testing cefiderocol has not been systematically assessed. Eventually, we decided to assume a MIC = 2 for our strains of DTR-AB, equal to the PK-PD susceptibility breakpoint suggested by EUCAST [13]. According to this assumption, ID1 and ID3 would have had *f*C_trough_/MIC ≥12 (Table 2). Furthermore, the presence of a *f*C_trough_/MIC ≥ 4 has been suggested recently as an effective target to achieve microbiological cure and reduce the emergence of multidrug resistance while treating DTR-AB [25]. ID2 had *f*C_trough_/MIC = 3.0. Based on the determination from ID1 and ID3, the ID2 cefiderocol *f*C_trough_ would have likely been higher during the acute phase of AKI, and we can speculate that when AKI is resolving, critically ill patients could benefit from a maximum dosage of 2 g q6 h.

Nevertheless, ID2 had an approximate theoretical T > MIC of 12 h, while in ID1 and ID3, it was >24 h, meaning an abundant coverage with a q8 h schedule. Thus, despite this initial renal dysfunction, the adoption of the lower dose of 0.75 g/12 h suggested by Katsube et al. for CVVH and considered as a standard dose adjustment in end-stage renal disease would have led to a clinically significant suboptimal exposure in ID2 [9].

Considering the hydric balance, this is unlikely to affect these PK data significantly because it was in a range of −1.6–1.0 L, theoretically accounting for a variation of between −8.8% and +5.5% in the mean volume of distribution of 18 L reported in adults [9].

Although we cannot make inferences, given the low number of patients studied, the microbiological cure was observed in all three patients described here. The high *f*C_trough_/MIC described in our patients could have helped achieve microbiological eradication even in challenging infections, such as VAP, in which optimal antibiotic lung concentrations are difficult to achieve, although larger studies are warranted. Nonetheless, we only speculated about the protein binding of cefiderocol, and the precise *f*C_trough_ concentration has not been calculated in this case series. Recently, it has been demonstrated that following increasing exposure to cefiderocol, there is the possibility of selection of some subpopulations of DTR-AB-harboring cefiderocol resistance, a phenomenon known as heteroresistance [25]. 

A dosing regimen of 2g q8h in patients undergoing CVVH has led to *f*C_trough_/MIC > 12 in two patients with AKI and to *f*C_trough_/MIC = 2.9 in one patient recovering from AKI. Despite the limit of MIC simulation, our data showed an effective achievement of *f*C_trough_/MIC that could be able to overcome the possibility of heteroresistance. The antibiotic concentration that enables the prevention of this phenomenon is called the Mutant Prevention Concentration (MPC). Thus, the therapeutic effort should be directed to keeping drug concentrations above MPC in order to restrict the emergence of MDR strains and achieve the clinical cure. ID2 presented lower-than-expected fCtrough/MIC, but we speculate that if the plasmatic concentration had been assessed a few days earlier during the acute phase of AKI, we would have expected fCtrough/MIC of a similar magnitude to those observed in ID1 and ID3. Moreover, it is important to consider that, considering the t1/2 2.0 h in this patient, the estimated f T < 4−6 × MIC was about 2 h over the day, meaning that the percentage of fT > 4−6 × MIC was nearly 92%.

Considering these factors, we conclude that even in critically ill patients with severe infections and AKI, the cefiderocol dosing of 2 g q8 h led to very high plasma exposure. No severe adverse events were observed in our three patients, but one developed maculopapular skin rash during cefiderocol treatment, which resolved after cefiderocol suspension and antihistaminic treatment. Given the few therapeutic options against DTR-AB and the reduced cefiderocol penetration into ELF, TDM results may be the option to enhance the dosing interval and achieve clinical cure.

Further data and larger studies are needed to explore the possibility of reduced dosing to attain the optimal target of 100% fT > 4−6 × MIC in a specific population, such as patients treated with CVVH.

All three patients achieved microbiological cure, even in difficult-to-treat infections, such as VAP, but the highly variable *f*C_trough_/MIC confirms the possible additional value of TDM in guiding the antibiotic treatment, especially when facing challenging infections in specific populations, such as patients undergoing CVVH with critical illness.

## 4. Materials and Methods

We described, retrospectively, the clinical course and PK characteristics of three critically ill patients undergoing CVVH treated with cefiderocol administered under compassionate use at University Hospital Città della Salute e della Scienza, Turin, Italy. Species identification was performed by matrix-assisted laser desorption ionization-time of fight mass spectrometry (MALDI-TOF MS) (Bruker DALTONIK GmbH, Bremen, Germany). The susceptibility to levofloxacin, meropenem, trimethoprim/sulfamethoxazole (TMP/SMX) and aminoglycosides (gentamycin and amikacin) was determined by a commercially available microdilution assay (Panel NMDR, Microscan WalkAway 96 Plus, Beckman Coulter, Switzerland), according to the manufacturer’s instructions. Colistin susceptibility was determined using the reference broth microdilution method. Cefiderocol antimicrobial susceptibility was determined using lyophilized panels (SensititreTM, ThermoFisher Scientific, Waltham, MA, USA) that proved to be comparable to the reference broth microdilution [12]. To confirm cefiderocol resistance with the microdilution method, a disc diffusion method (LiofilchemVR, Roseto degli Abruzzi, Italy) was adopted on a standard Mueller–Hinton agar and incubated for 18–24 h at 35 ± 2 °C, as recommended by EUCAST.

Cefiderocol (Fetcroja^®^, Shionogi & Co., Ltd., Osaka, Japan) was reconstituted from vials with sterile water, following the manufacturer’s instructions.

At the time of data collection, susceptibility rates for cefiderocol against DTR-AB were not assessed in our Centre, thus we chose to adopt the PK-PD MIC breakpoint of S = 2 mg/L, as suggested by EUCAST [13].

For each patient, samples were collected on day 7 since the start of treatment, corresponding to the 22nd dose of cefiderocol treatment administered to the dosage of 2 g every 8 h infused over 3 h during CVVH. Plasma samples for assessing plasma cefiderocol trough concentrations (Cmin) were collected 30 min before one of the daily administrations, following achievement of steady-state conditions. The other samples were collected at the end of infusion, 2 h and 4 h after the end of infusion. Blood samples collected for PK analysis were obtained from a different intravenous line adopted for cefiderocol infusion. Blood samples were stored frozen at −80 °C and shipped on dry ice to the Laboratory of Clinical Pharmacology and Pharmacogenetics, “Amedeo di Savoia” Hospital, Turin, for the quantification of cefiderocol plasma concentration through a validated UltraHigh Performance Liquid Chromatography coupled with tandem mass spectrometry (UHPLC-MS/MS) assay, using the KitSystem Antibiotics^®^ analytical kit (CoQua Lab, Turin, Italy). The sample preparation protocol consisted of a protein precipitation of 50 µL of plasma with 150 µL of precipitating solution. The supernatant was then diluted and injected in the chromatographic system (LX−50 UHPLC with QSight 220^®^, Perkin Elmer, Milan, Italy). The lower (LLOQ) and upper (ULOQ) limits of quantification for cefiderocol quantification were 3.75 mg/L and 120 mg/L, respectively. The PK results were then analyzed by non-compartmental analysis (NCA) with Phoenix WinNonLin^®^ software (Certara, Princeton, NJ, USA), with a “linear-up/log-down” approach, in order to obtain the areas under the concentration-time curve (AUC) and half-life (t1/2) parameters. T4 was considered as a proxy of trough concentration (Ctrough), while T1 was considered as the peak concentration (Cmax). With regard to PK/PD considerations, *f*C_trough_ were calculated considering 58% plasma protein binding and the observed MIC from susceptibility tests in order to verify the achievement of 100% of time (T) fC > 4−6 × MIC. On the day of PK analysis, the CVVH settings of patients were as described in Table 3. Written informed consent was not obtained due to the critical illness of the patients, which required invasive mechanical ventilation; thus, the principle of urgency was applied. CVVH was conducted with a heparinized circuit with the following settings: blood flow rate 200 mL/min, dialysate flow rate 200 mL/min. The effluent flow rate was 2300 mL and 2600 mL, respectively, for patients 1 and 2 on the day of sampling. The hydric balance was negative for both patients (−342 mL and −180 mL on the day of sampling, respectively). The settings for ID3 were as follows: blood flow 200 mL/min, hydric balance −100 mL, effluent flow rate −3600 mL on the day of sampling.

## Figures and Tables

**Table 1 antibiotics-11-01830-t001:** Susceptibility of isolates of *A. baumannii*.

	Isolate	Levofloxacin	Meropenem	TMP/ SMX ^3^	Aminoglycosides	Colistin
Patient 1	*A. baumannii*	>2(R)	>8(R)	>4/76(R)	>16(R) ^1^	0.5(S)
Patient 2	*A. baumannii*	>2(R)	>8(R)	>4/76(R)	>16(R) ^1^	1(S)
Patient 3	*A. baumannii*	>1(R)	>32(R)	>4/76(R)	>4(R) ^2^	0.5(S)

^1^ Amikacin; ^2^ Gentamycin; ^3^ TMP/SMX trimethoprim/sulfamethoxazole; R: resistant; S: susceptible.

**Table 2 antibiotics-11-01830-t002:** Summary of the observed pharmacokinetic and pharmacokinetic/pharmacodynamic parameters of Cefiderocol.

Patient	Ctrough (mg/L)	*f*C_trough_ (mg/L)	*f*C_trough_/MIC	Cmax (ng/mL)	T_1/2_	AUC_0–8 h_ (mg/L*h)
ID1	57.95	24.34	12.2	104.94	7.6	643.90
ID2	14.17	5.95	3.0	64.87	2.0	272.11
ID3	71.66	30.10	15.0	117.40	9.6	772.66

Ctrough: trough concentration; *f*C_trough_: free trough concentration; MIC: minimum inhibitory concentration; Cmax: maximum plasma concentration; T1/2: half-life; AUC: area under the curve.

**Table 3 antibiotics-11-01830-t003:** Mean clinical and demographic characteristics of ICU patients undergoing continuous venovenous hemofiltration treated with cefiderocol.

ID	Age, Sex (Y; M/F)	Weight (Kg)	Comorbidities	Albumin (mg/dL)	SOFA	APACHE II	Type of Infection	Renal Support	Cytokines Filter	Residual Diuresis	Dehydration	Efflux Rate (mL)	30-Days Mortality
1	56, M	85	Lung Tx	2.4	13	17	VAP	CVVH	CYTOSORB^®^	20 mL	−342 mL/h	2300	no
2	71, M	83	SARS-CoV-2 Pneumonia	2.4	11	8	BSI	CVVH	EMiC^®^2	25 mL	−180 mL/h	2600	no
3	45, M	110	HTA	5.0	10	21	VAP, BSI	CVVH	EMiC^®^2	0 mL	−100 mL/h	3600	no

Sofa: SOFA: Sequential Organ Failure Assessment; APACHE II: Acute Physiology and Chronic Health Disease Classification System II; BSI: bloodstream infection; VAP: ventilator associated pneumonia; CVVH: Continuous venovenous haemofiltration; HTA: arterial hypertension; Tx: transplant.

## Data Availability

Data available on request from the authors.

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
