# Peer review of "Pharmacokinetic of Cefiderocol in Critically Ill Patients Receiving Renal Replacement Therapy: A Case Series"

_antibiotics, 2022, doi:10.3390/antibiotics11121830_

Round 1

Reviewer 1 Report

The study is very interesting since it is about a novel antibiotic and its use in critical patients.

Authors performed cefiderocol TDM in patients with MDR infections and RRT showing adequate dosing regimens. Although they reported few cases, I believe the results would be useful for further studies.

The English is great, and it only needs minor spell checks and typos correction.

In addition, I have the following concerns:

·       Line 38: Use plural form of “infections”

·       Line 41: Insert a comma “…combination, and carbapenems”

·       Line 44: “first cephalosporin to utilize” please rephrase

·       Line 62: Better to say “Here we report…”

·       Line 84: I suggest deleting the word “needs”, the meaning would be the same.

·       Line 105: “but worsening gas exchange developed after…” please rephrase

·       Case 2: You stated (Line 115) that blood cultures resulted positive for Enterococcus faecalis and E. faecium, what antibiotic therapy did you administer for those germs? Did you repeat blood cultures (you stated that you repeated only BAL)? Please clarify and add missing information.

Have you adjusted fosfomycin dosage for renal function?

·       Line 141: Please write the meaning of IOT or the full form

·       Line 147: Was the Staph. aureus MS? Please clarify

·       Table 1: Please adjust the caption (some missing “:” and it lacks some acronyms such as HTA)

·       Line 188: Please add a reference here

·       Line 199: add unit of measure after 0.42

·       Please add few lines explaining the choice of monotherapy rather than combination to treat DTR-AB. Do you think cefiderocol could be used alone against Acinetobacter? Furthermore, discuss the possible role of fosfomycin, which you administered to ID2, as partner drug for cefiderocol in AB infections (10.3390/antibiotics10060652)

·       In addition to improve discussion section I suggest reading and adding this paper 10.1128/spectrum.02347-22.

Kind regards

Author Response

We would like to thank the reviewer for careful and thorough reading of this manuscript and for the useful comments. Hereafter you 'll find our detailed response to comments. 

- Following your suggestion, all spelling and grammatical errors pointed out by the reviewers have been corrected on lines 38-41-62-84.

 - Line 44: “first cephalosporin to utilize” please rephrase

The sentence has been changed as follows: Cefiderocol is novel chlorocatechol-substituted siderophore cephalosporin known to form an iron-chelating complex to gain access to Gram-negatives membrane, showing enhanced in vitro activity against DTR Enterobacterales and NFGNBs, including strains harboring metallo-β-enzymes.

- Line 105: “but worsening gas exchange developed after…” please rephrase.

Line 105 has been rephased as follows: Oxygen via nasal cannula was started, but acute respiratory failure developed after 1 day, requiring ventilation with HCPAP.

- Case 2: You stated (Line 115) that blood cultures resulted positive for Enterococcus faecalis and E. faecium, what antibiotic therapy did you administer for those germs? Did you repeat blood cultures (you stated that you repeated only BAL)? Please clarify and add missing information.

Thank you for pointing this out. You have raised an important point here, that given the limited number of words we did not assessed in the Manuscript. Fosfomycin has a unique mechanism of action, characterized by bactericidal activity against Gram-positive and Gram-negative pathogens both. There is a robust literature supporting high in vitro activity against a wide selection of bacteria including methicillin-resistant Staphylococcus aureus (MRSA), vancomycin-resistant enterococci (VRE), multidrug-resistant (MDR) Enterobacteriaceae, and some isolates of MDR Pseudomonas aeruginosa (DOI: https://doi-org.bvsp.idm.oclc.org/10.1128/AAC.02099-12; Diagn Microbiol Infect Dis, 42 (2002), pp. 269-271). Furthermore, fosfomycin demonstrated an interesting synergistic activity when used in combination with a wide selection of antimicrobials(DOI: https://doi-org.bvsp.idm.oclc.org/10.1128/AAC.02099-12). Unfortunately, there are not RCT or clinical trial that have assessed superiority or non-inferiority of Fosfomycin as monotherapy vs best available treatment against vancomycin resistant E.faecium (VRE) and fosfomcyin has been used mainly within combination-regimen. Nevertheless, fosfomycin has become a valuable combination partner in the treatment of MDR gram positives and gram-negative, supported by its advantageous pharmacokinetic properties. In addition, the high dosing of fosfomycin we chose (24g daily) was likely to achieve effective concentration in the treatment of bactermia secondary to urinary tract infection from E.faecalis and E.faecium. Furthermore, in this particular case we chose fosfomycin-containing regimen for the encouraging data of clinical efficacy against DTR-AB which was the main issue in our patient. Given the clinical response, with rapid resolution of septic shock we did not repeat surveillance blood cultures, but we repeat BAL culture, that resulted negative, as reported on line 130. This represent the test of cure of VAP.

Following your suggestion, this sentence has now been corrected as following on lines 124-126 “Antibiotic treatment was changed to cefiderocol 2g every 8 hours and iv fosfomycin 4g every 6 hours was added for its intrinsic activity against Enterococcus spp. and its favourable PK/PD behavior in the lung tissue and synergistic effect when used whitin a combination treatment”.

- Have you adjusted fosfomycin dosage for renal function?

We very much appreciate this helpful comment that is of great interest. There is a lack of data assessing the optimal fosfomycin dose in patients undergoing CVVH. Previously has been reported that fosfomycin at the dosage of 16 g/day (Antimicrob Agents Chemother. 2020;65:e01375-e1420;  Eur J Hosp Pharm Sci Pract. 2018;25:e115–9.) could achieve effective concentration in the presence of pathogens with high MICs, nevertheless, considering the criticall illness, the septic shock and the need to penetrate in a difficult site as the lung we chose to administer the maximum daily dose of 24g, suggested for MDR organisms. In addition the CVVH helped to protect the patients from electrolyte disorders that occur during fosfomycin treatment.

- Line 141: Please write the meaning of IOT or the full form

Following your suggestion, the significance of IOT has been added.

- Line 147: Was the Staph. aureus MS? Please clarify

Agree. We have, accordingly, revised the sentence adding the methicillin susceptibility of the S.aureus.

- Table 1: Please adjust the caption (some missing “:” and it lacks some acronyms such as HTA)-

We have, accordingly, added the acronyms. We don’t find any missing in the table.

Line 188: Please add a reference here

We have, accordingly, added the references.

-  Line 199: add unit of measure after 0.42

There is no unit of measure for unbound fraction because is the ratio ratio of the free concentration and total concentration, otherwise reported as 42%.

-Please add few lines explaining the choice of monotherapy rather than combination to treat DTR-AB. Do you think cefiderocol could be used alone against Acinetobacter? Furthermore, discuss the possible role of fosfomycin, which you administered to ID2, as partner drug for cefiderocol in AB infections (10.3390/antibiotics10060652).

Thank you for this suggestion. It would have been interesting to explore this aspect. I’ve explained earlier the reasons we chose fosfomycin in this particular case. However, in the case of our study, it seems slightly out of scope to explain the combination/monotherapy because the main focus of the manuscript is the plasmatic concentration of cefiderocol in critically ill patients undergoing CVVH but we have added a short explanation on lines 127-130 and the reference as you suggested [15].

- In addition to improve discussion section I suggest reading and adding this paper 10.1128/spectrum.02347-22.

We agree with this and have incorporated your suggestion throughout the manuscript from line 286. We rephrased as follows “Recently, it has been demonstrated that following increasing exposure to cefiderocol, the regrowth of some subpopulations of DTR-AB harboring cefiderocol resistance was noted, a phenomenon known as heteroresistance[34]. In the setting of high inoculum infections, such as VAP, the occurrence of cefiderocol resistance VAP is a major concern. In this setting the role of combination treatment with fosfomycin should be studied in clinical trials.”

Reviewer 2 Report

Dear authors

Please find my comments below

1) Line 320: The plasma samples collection wording is little bit confusing to -   readers. Please address.

2) Typically blood samples were treated on the spot to get plasma. Are you guys storing and transporting blood samples or plasma samples? if blood samples were stored @-80C., then for how long?

3) What kind of method validation guidelines/ parameters were followed for UHPLC-MS/MS?

4) For species identification you have used MALDI MS. Please specify what kind of species.

Author Response

Reviewer 2

Comments and Suggestions for Authors

Dear authors

Please find my comments below

1) Line 320: The plasma samples collection wording is little bit confusing to -   readers. Please address.

2) Typically blood samples were treated on the spot to get plasma. Are you guys storing and transporting blood samples or plasma samples? if blood samples were stored @-80C., then for how long?

3) What kind of method validation guidelines/ parameters were followed for UHPLC-MS/MS?

4) For species identification you have used MALDI MS. Please specify what kind of species.

Submission Date

06 November 2022

Date of this review

24 Nov 2022 05:54:15

© 1996-2022 MDPI (Basel, Switzerland) unless otherwise stated

Disclaimer Terms and Conditions Privacy Policy

Response to reviewer 2

We are very grateful for the reviews you provided to manuscript. Please see below, your considerations, in black, and our detailed response in red.

1) Line 320: The plasma samples collection wording is little bit confusing to -   readers. Please address.

- We thank the reviewer for this comment and we totally agree that explanation is a bit confusing. We prepared the same protocol for patients undergoing CVVH or intermittent dialysis so used a particular numeric order. To facilitate interpretation of the timing of sampling time , we have slightly reworded as follows: “Plasma samples for assessing plasma cefiderocol trough concentrations (Cmin) were collected 30 min before one of the daily administrations following achieving steady-state conditions. The other samples were collected at the end of infusion, 2 hours and 4 hours after the end of infusion”.

2) Typically blood samples were treated on the spot to get plasma. Are you guys storing and transporting blood samples or plasma samples? if blood samples were stored @-80C., then for how long?

- Thanks for the request of clarification. I’ll explain in brief our method for specimenscprocessing. After blood collection within EDTA tubes,

the whole blood was temporally conserved in a refrigerator one hour maximum. Then the sample was centrifuged speed of 1400 RCF for 10 minutes. Centrifuge temperature: +4°C. Plasma was drawn into the plasma sample cryotubes and storage at -80°C that allows an optimal conservation during several months. Our samples have been analyzed within 1 month after storage.

3) What kind of method validation guidelines/ parameters were followed for UHPLC-MS/MS?

-Quantification has been made with a kit UHPLC-MS/MS (KIT-SYSTEM Antibiotics, CoQua Lab, Torino, Italy) CE-IVD marked and validated in accordance with EMA and FDA guidelines. We followed these guidelines:

1EMA, Guideline on bioanalytical method validation. http://www.ema.europa.eu/docs/en_GB/document_library/Scientific_guideline/2011/08/WC500109686.pdf, 2011 (accessed 04-03-2021.).

2FDA, Guidance for Industry: Bioanalytical Method Validation. https://www.fda.gov/files/drugs/published/Bioanalytical-Method-Validation-Guidance-for-Industry.pdf, 2013 (accessed 04/03/2021.).

4) For species identification you have used MALDI MS. Please specify what kind of species.

- Matrix assisted laser desorption ionization-time of flight mass spectrometry (MALDI-TOF MS) has emerged as a potential tool for microbial identification. Matching the masses of biomarkers of organisms. It allows an early  identification of microorganisms (bacteria in our manuscript) to up to the level of species.